

# Comparative analysis of four *Zantedeschia* chloroplast genomes: expansion and contraction of the IR region, phylogenetic analyses and SSR genetic diversity assessment

Shuilian He[1,*], Yang Yang[2,*], Ziwei Li[1], Xuejiao Wang[1], Yanbing Guo[1] and Hongzhi Wu[3]

[1] College of Horticulture and Landscape, Yunnan Agriculture University, Kunming, Yunnan, China
[2] College of Science, Yunnan Agriculture University, Kunming, Yunnan, China
[3] College of horticulture and landscape, Yunnan Agricultural University, Kunming, Yunnan, China
[*] These authors contributed equally to this work.

## ABSTRACT

The horticulturally important genus *Zantedeschia* (Araceae) comprises eight species of herbaceous perennials. We sequenced, assembled and analyzed the chloroplast (cp) genomes of four species of *Zantedeschia* (*Z. aethiopica*, *Z. odorata*, *Z. elliottiana*, and *Z. rehmannii*) to investigate the structure of the cp genome in the genus. According to our results, the cp genome of *Zantedeschia* ranges in size from 169,065 bp (*Z. aethiopica*) to 175,906 bp (*Z. elliottiana*). We identified a total of 112 unique genes, including 78 protein-coding genes, 30 transfer RNA (tRNA) genes and four ribosomal RNA (rRNA) genes. Comparison of our results with cp genomes from other species in the Araceae suggests that the relatively large sizes of the *Zantedeschia* cp genomes may result from inverted repeats (IR) region expansion. The sampled *Zantedeschia* species formed a monophylogenetic clade in our phylogenetic analysis. Furthermore, the long single copy (LSC) and short single copy (SSC) regions in *Zantedeschia* are more divergent than the IR regions in the same genus, and non-coding regions showed generally higher divergence than coding regions. We identified a total of 410 cpSSR sites from the four *Zantedeschia* species studied. Genetic diversity analyses based on four polymorphic SSR markers from 134 cultivars of *Zantedeschia* suggested that high genetic diversity ($I = 0.934$; $Ne = 2.371$) is present in the *Zantedeschia* cultivars. High genetic polymorphism from the cpSSR region suggests that cpSSR could be an effective tool for genetic diversity assessment and identification of *Zantedeschia* varieties.

# INTRODUCTION

The genus *Zantedeschia* Spreng. (Trib. Richardieae, Araceae) had originally an entirely northeastern and southern African distribution. However, following introduction to Europe as ornamental plants in the seventeenth century, various species became widely

Corresponding author
Hongzhi Wu, 1994061@ynau.edu.cn

naturalized across Europe, America, Oceania and Asia (*Cruzcastillo, Mendozaramirez & Torreslima, 2001*). Two sections, *Zantedeschia* and *Aestivae*, are currently recognized in the genus *Zantedeschia*. Species in section *Zantedeschia*, *Z. aethiopica* and *Z. odorata* (*Singh, Wyk & Baijnath, 1996*), can be recognized by the rhizomatous tuber and white flowers. Species in section *Aestivae*, however, have colorful (not white) flowers and discoid tubers (*Singh, Wyk & Baijnath, 1996*; *Wright & Burge, 2000*). Many attractive and colorful hybrids between species in section *Aestivae* have been artificially produced, the majority between *Z. albomaculata, Z. elliotiana, Z. rehmannii* and *Z. pentlandii* (*Snijder et al., 2004a*). *Zantedeschia* hybrids have subsequently become one of the most popular horticultural crops worldwide, in high demand as cut flowers, potted plants and flower baskets, as well as for use in flower beds. The genus *Zantedeschia* is also of horticultural interest, however, F1 hybrids between sections *Zantedeschia* and *Aestivae* are invariably albino.

Traditional and polyploidization breeding, as well as resistance to soft rot, have been the main focuses for previous research into the genus *Zantedeschia* (*Snijder et al., 2004a*; *Snijder, Lindhout & Van Tuyl, 2004b*; *Wright & Triggs, 2009*; *Wright & Burge, 2000*; *Wright, Burge & Triggs, 2002*). Simple sequence repeats (SSRs), or microsatellites, are short tandem repeats of two to more nucleotides in DNA sequences. The number of repeats is highly variable, whereas the regions of DNA flanking SSRs are highly conserved (*Davierwala et al., 2000*; *Gur-Arie et al., 2000*). SSR markers are polymerase chain reaction (PCR) -based, abundant, codominant, highly reproducible, and are distributed evenly across eukaryotic genomes (*Powell et al., 1996*). SSRs are widely used molecular markers to study genetic diversity, population structure, genetic mapping, phylogenetic studies, cultivar identification and marker-assisted selection (*Potter et al., 2015*). A total of 43 novel EST-derived simple sequence repeat (SSR) markers have been identified in *Zantedeschia* by *Wei et al. (2012)*, however, apart from this, the genetics and genomics of the genus *Zantedeschia*, which are of great importance in plant breeding, have received little research attention. We therefore recommend that further genomic resources from *Zantedeschia* should be developed as tools to assist molecular breeding research in this genus. Our study focuses on four species of *Zantedeschia*, two from section *Zantedeschia* and two from section *Aestivae*. The aim of the study was to sequence, assemble and analyze the cp genome in *Zantedeschia*, to investigate any common characteristics or differences between the studied species and also to develop SSR markers in the *Zantedeschia* cp genome.

Photosysnthetic fixation of carbon in plants takes place in the chloroplasts, and is a primary function of these organelles. Chloroplasts have their own genome, as do mitochondria and it has been suggested that they were originally free-living cyanobacterium-like cells engulfed by ancient eukaryotic cells in an endosymbiotic relationship (*Raven & Allen, 2003*). The cp genome is usually represented as a circular molecule, and has a conserved quadripartite structure comprising the small single copy (SSC) and large single copy (LSC) regions, separated by two copies of an inverted repeat (IR) region. Chloroplast genomes have a highly conserved gene content, and most land plants have a nearly collinear gene sequence (*Jansen et al., 2005*). Due to their lack of recombination, their compact size and their maternal inheritance (*Birky, 2001*), cp genomes

are considered to be useful DNA sequences for plant genetic diversity assessment, plant identification and phylogenetic studies.

We investigated the cp genomes from four species of *Zantedeschia*. Genomes were sequenced, assembled, annotated and mined for the presence of SSR markers using Illumina sequencing technology. We also made comparative sequence analysis studies of the cp sequences from our study species. These results are publicly available as a genetic resource for the study of *Zantedeschia* species, and it is our hope that they will provide a valuable resource for future genetic and phylogenetic studies into this important genus.

## MATERIALS & METHODS

### Plant material, DNA sequencing and cp genome assembly

Plant material of *Z. aethiopica* was collected from South Africa directly and has been planted in Kunming more than 30 years. *Z.odorata, Z. elliottiana, and Z. rehmannii* were collected from Netherlands. Total genomic DNA of the four species of *Zantedeschia* was extracted from the fresh leaves of tissue culture seedlings using a modified CTAB extraction protocol based on *Doyle & Doyle (1987)*. Sequencing of the genomic DNA was performed using an Illumina Hiseq2000 (Illumina, CA, USA). Low quality reads were filtered out before *de novo* assembly of the cp genomes, and the resulting clean reads were assembled using the GetOrganelle pipeline (https://github.com/Kinggerm/GetOrganelle). A reference genome *Colocasia esculenta* (JN105689) was used to check the contigs, using BLAST (https://blast.ncbi.nlm.nih.gov/), and the aligned contigs were then oriented according to the reference genome.

### Gene annotation and sequence analysis

The CpGAVAS pipeline (*Liu et al., 2012*) was used to annotate the genome and start/stop codons and intron/exon boundaries were adjusted in Geneious 8.1 (*Kearse et al., 2012*). The tRNA was identified using tRNAscan-SE v2.0 (*Lowe & Chan, 2016*), and sequence data were subsequently deposited in GenBank. The online tool OGDraw v1.2 (http://ogdraw.mpimp-golm.mpg.de/, *Lohse, Drechsel & Bock, 2007*) was used to generate a physical map of the genome.

### Structure of Genome and Genome comparison

Pairwise sequence alignments of cp genomes were performed in MUMer (*Kurtz et al., 2004*).The complete cp genomes of the four species were then compared using mVISTA (*Mayor et al., 2000*) with the shuffle-LAGAN model Codon usage bias (RSCU) was calculated using MEGA v7.0 (*Kumar et al., 2008*). Chloroplast genome sequences of the four species were aligned using MAFFT (*Katoh & Standley, 2013*) in Geneious 8.1 (*Kearse et al., 2012*). Insertion/deletion polymorphisms (indels) were then identified using DnaSP version 5.1 with the cp genome of *Z. aethiopica* as a reference (*Librado & Rozas, 2009*). Single nucleotide polymorphisms (SNPs), defined as variations in a single nucleotide that occur at specific positions in the genome, were called using a custom Python script (https://www.biostars.org/p/119214/).

## Phylogenetic analysis

Chloroplast genome sequences of 11 species from Araceae and an outgroup (*Zea mays*, Poaceae) were downloaded from GenBank and an alignment with the four *Zantedeschia* cp genome sequences from our study was built using MAFFT (*Katoh & Standley, 2013*) in Geneious 8.1 (*Kearse et al., 2012*). In order to investigate the phylogenetic placement of the genus *Zantedeschia* within the Araceae, a maximum likelihood tree was reconstructed in RaxML version 8.2.11 (*Stamatakis, 2014*). Tree robustness was assessed using 1000 replicates of rapid four bootstrapping with the GTR+GAMMA substitution model.

## Simple Sequence Repeats (SSRs)

SSR markers present in the *Zantedeschia* cp genome were found using Phobos v3.3.12 (*Leese, Mayer & Held, 2008*) and SSRHunter (*Li & Wan, 2005*). Both these programs search for repeats using a recursive algorithm. We set the minimum number of repeats of mono-, di-, tri-, tetra-, penta-, and hexa-nucleutide repeats to 10, 5, 4, 3, 3 and 3 respectively. The inverted repeat region IRa was not considered in our SSR analysis.

We subsequently selected four SSRs motifs to investigate genetic diversity in *Zantedeschia*. A total of 134 cultivars from genus *Zantedeschia* were sampled. The experimented 134 cultivars include some local cultivars, but most of them are collected from Netherlands, the United States, New Zealand, and Taiwan for production and refreshed by tissue culture every 3 years. Genetic diversity was investigated by calculating several indices: the number of alleles per locus ($Na$); the number of fffective alleles ($Ne$); Shannon's information index ($I$) and polymorphism information content (PIC). $Na$, $Ne$, I were calculated using GenALEx v. 6.4 (*Peakall & Smouse, 2006*). PIC was calculated using PowerMarker 3.25 (*Liu & Muse, 2006*).

## RESULTS

### Characteristics of *Zantedeschia* cp genomes

After assembly and annotation, the four *Zantedeschia* cp genomes obtained in this study were submitted to the NCBI database (accession numbers MH743153–MH743155 and MG432242). The cp genomes of these *Zantedeschia* species ranged in length from 169,065 bp (*Z. aethiopica*) to 175,906 bp (*Z. elliottiana*), and, as expected, the cp genomes of all four species were found to contain both the large and small single-copy regions, separated by a pair of inverted repeat regions (Fig. 1 & Fig. S1, Table 1). A total of 139 genes, of which 112 were unique, were identified, including 93 (78 unique) protein-coding genes, 38 (30 unique) transfer RNA (tRNA) genes and eight (four unique) ribosomal RNA (rRNA) genes (Table 2).

Interestingly, although the four species belong to a single genus, differences in gene content can nevertheless be seen. *Z. elliottiana* had the largest number of genes (139). *Z. rehmannii* had 138 genes, differing from *Z. elliottiana* only in a single copy of rps19. *Z. odorata* had 134 genes, and differs from *Z. elliottiana* in having only single copies of *ndhE*, *ndhG*, *rps19*, *trnH-GUG* and *trnV-UAC*. Of all the species we studied, the cp genome of *Z. aethiopica* had the fewest of genes (131), and differed from *Z. elliottiana* in single copies of *ndhA*, *ndhE*, *ndhG*, *ndhH*, *ndhI*, *rps19*, *trnH-GUG* and *trnV-UAC* (Table 2).

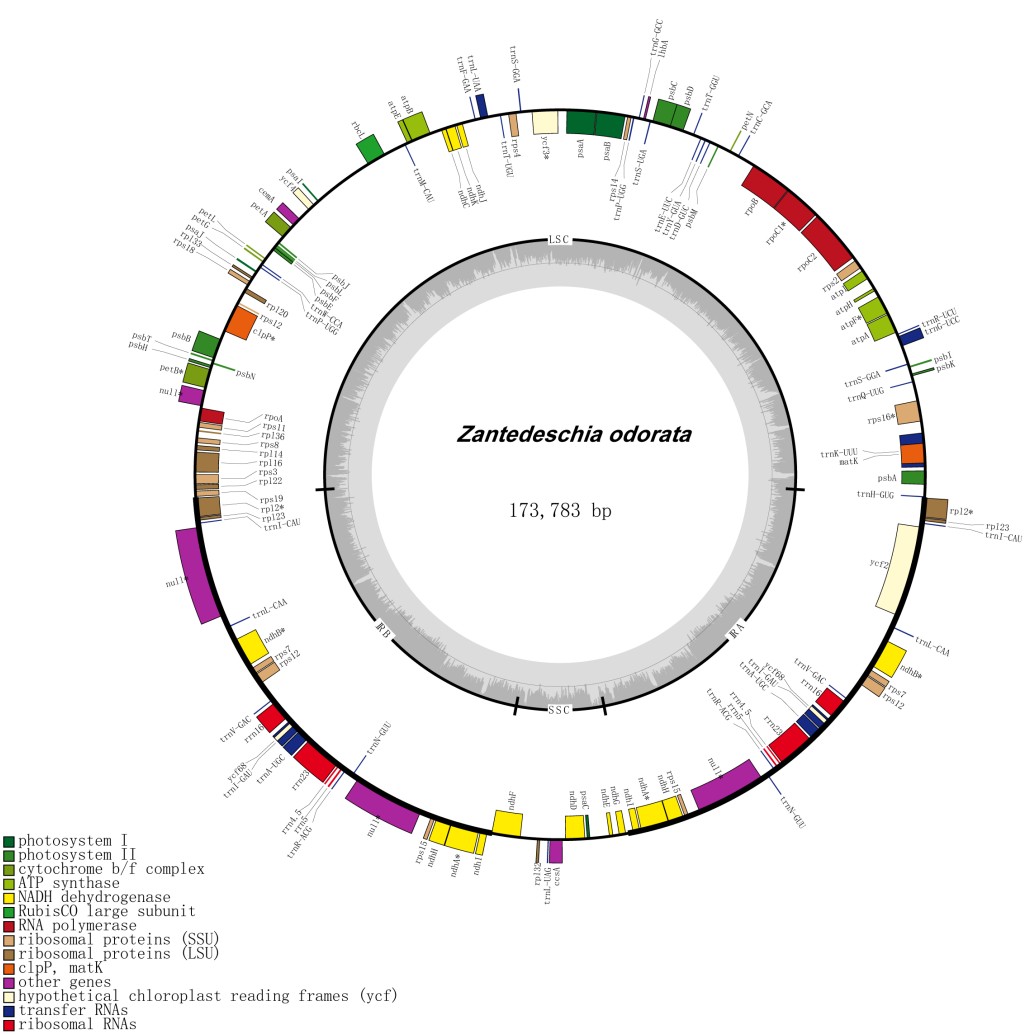

photosystem I
photosystem II
cytochrome b/f complex
ATP synthase
NADH dehydrogenase
RubisCO large subunit
RNA polymerase
ribosomal proteins (SSU)
ribosomal proteins (LSU)
clpP, matK
other genes
hypothetical chloroplast reading frames (ycf)
transfer RNAs
ribosomal RNAs

**Figure 1   Gene map of chloroplast genome of *Z. odorata*.**

The two species from section *Aestivae* (*Z. elliottiana*; *Z. rehmannii*) have larger cp genomes that those in section *Zantedeschia* (*Z. aethiopica* and *Z. odorata*), and the number of different protein-coding genes and tRNA genes is also higher in the cp genomes from plants in section *Aestivae*. Moreover, the size of the IR regions in species from section *Aestivae* was also larger than those in section *Zantedeschia*. Unlike the other studied species, which had no pseudogene, *Z. odorata* had two copies of the pseudogene Ψ*ycf68*.

The nucleotide composition of the *Zantedeschia* cp genome was asymmetric, with an overall GC content ranging from 35.3% to 35.6%, which is similar to other species in the Araceae (*Tian et al., 2018*). The largest GC content ratio was observed in the IR region (37.5%–39.0%), and the smallest in the SSC region (28.2%–29.6%). All four of our study species showed the same trend (the GC content of the LSC and SSC regions was lower than that of the IR regions), which may be due to the tRNA genes and rRNA genes generally having fewer AT nucleotides (*Chen et al., 2015*; *Meng et al., 2018*; *Zhou et al., 2017*).

**Table 1   The basic characteristics of chloroplast genomes of four *Zantedeschia* species.**

| Characteristics | *Z. elliottiana* | *Z. rehmannii* | *Z. aethiopica* | *Z. odorata* |
|---|---|---|---|---|
| GenBank numbers | MH743153 | MH743154 | MH743155 | MG432242 |
| Total cp genome size/bp | 175,906 | 175,067 | 169,065 | 173,783 |
| LSC size/bp | 88,584 | 90,020 | 89,695 | 90,322 |
| IR size /bp | 39,445 | 38,354 | 32,331 | 36,549 |
| SSC size /bp | 8,432 | 8,338 | 14,715 | 10,363 |
| Total number of genes | 139 | 138 | 131 | 134 |
| Number of different protein-coding genes | 93 | 92 | 87 | 90 |
| Number of different tRNA genes | 38 | 38 | 37 | 36 |
| Number of different rRNA genes | 8 | 8 | 8 | 8 |
| Number of gene in IR region | 54 | 52 | 40 | 46 |
| Number of pseudogene | 0 | 0 | 0 | 2 |
| GC content (%) | 35.4 | 35.6 | 35.6 | 35.3 |
| GC content of LSC (%) | 34.2 | 34.4 | 34.1 | 33.7 |
| GC content of IR (%) | 37.5 | 37.7 | 39.0 | 38.2 |
| GC content of SSC (%) | 28.7 | 29.1 | 29.6 | 28.2 |

Introns are non-coding sequences within genes, and they play a very important role in the regulation of gene expression (*Jiang et al., 2017*). Introns are known to accumulate more mutations than functional genes, and because of this are used extensively in phylogenetic and population genetics studies (*Xu, 2003*). We investigated 13 intron-containing genes in the species *Z. aethiopica*: 11 genes (*atpF*, *ndhA*, *ndhB*, *petB*, *petD*, *rpl16*, *rpl2*, *rpoC1*, *rps16*, *rps18*, *ycf68*) contained only one intron, while two genes (*clpP*, *ycf3*) contained two introns. *Z. elliottiana* and *Z. rehmannii* each had 12 intron-containing genes (similar to *Z. aethiopica* but lacking *clpP*), and 11 intron-containing genes were found in *Z. odorata* (as *Z. aethiopica* but lacking *rps18* and *ycf68*) (Table S1).

## Genome comparison

An extremely important topic in genomics is the comparative analysis of cp genomes (*Chen et al., 2012*; *Zhihai et al., 2016*). We performed multiple alignments between the four cp genomes generated in this study to characterize their divergence. The alignments were conducted in mVISTA, using *Z. odorata* as a reference (Fig. 2).

Unsurprisingly, we found that in our study species, the coding regions are more conserved than the non-coding regions. Furthermore, the LSC and SSC regions are more divergent from each other than the IR regions. Intergenetic spacers (including *trnH-psbA*, *trnK-rps16*, *rps16-psbK*, *trnT-trnL*, *rbcL-psaL*, *clpP-psbB*, *ycf1-trnL*, *trnL-ndhB* in the LSC regions and *psaC-ndhE*, *rps15-ycf1*, *trnL-ycf2* in the IR regions) were found to be the most divergent regions of the four cp genomes. Of the coding regions, the greatest divergence was found in *clpP*, *rpl16*, *rps19*, *ycf1* and *ycf2*. This is in agreement with the results from previous studies (*Park et al., 2017*; *Shen et al., 2017*; *Wu et al., 2017*), and suggests that these regions might evolve rapidly in the genus *Zantedeschia*.

**Table 2  Genes present in the *Zantedeschia* chloroplast genome.**

| Category | Gene groups | Name of genes |
|---|---|---|
| Self-replication | Large subunit of ribosomal proteins | *rpl2²*, *rpl14*, *rpl16*,*rpl20*, *rpl22*, *rpl23 ²*, *rpl32*, *rpl33*, *rpl36* |
| | Small subunit of ribosomal proteins | *rps2*, *rps3*, *rps4*, *rps7²*, *rps8*, *rps11*, *rps12²*, *rps14*, *rps15²*,*rps16*, *rps18*, *rps19 ²,b,c,a* |
| | DNA dependent RNA polymerase | *rpoA*, *rpoB*, *rpoC1*, *rpoC2* |
| | Ribosomal RNA genes | *rrn4.5²*, *rrn5²*, *rrn16 ²*, *rrn23²* |
| | Transfer RNA genes | *trnA(UGC)²*, *trnC(GCA)*, *trnD(GUC)*, *trnE(UUC)*, *trnF(GAA)*, *trnfM(CAU)*, *trnG(GCC)*, *trnG(UCC)*, *trnH(GUG)²,b,c*, *trnI(CAU)²*, *trnI(GAU)²*, *trnK(UUU)*, *trnL(CAA)²*, *trnL(UAA)*, *trnL(UAG)*, *trnM(CAU)*, *trnN(GUU)²*, *trnP(UGG)*, *trnQ(UUG)*, *trnR(ACG)²*, *trnR(UCU)*, *trnS(GCU)*, *trnS(GGA)*, *trnS(UGA)*, *trnT(GGU)*, *trnT(UGU)*, *trnV(GAC)²*, *trnV(UAC)\* d* *trnW(CCA)*, *trnY(GUA)* |
| Photosynthesis | NADH oxidoreductase | *ndhA²,b*, *ndhB²*, *ndhC*, *ndhD*, *ndhE²,b,c*, *ndhF*, *ndhG²,b,c* *ndhH ²,b*, *ndhI ²,b*, *ndhJ*, *ndhK* |
| | Photosystem I | *psaA*, *psaB*, *psaC*, *psaI*, *psaJ*,*ycf3*, *ycf4* |
| | Photosystem II | *psbA*, *psbB*, *psbC*, *psbD*, *psbE*, *psbF*, *psbH*, *psbI*, *psbJ*, *psbK*, *psbL*, *psbM*, *psbN*, *psbT*, *psbZ* |
| | Cytochrome b/f complex | *petA*, *petB*, *petD*, *petG*, *petL*, *petN* |
| | ATP synthase | *atpA*, *atpB*, *atpE*, *atpF*, *atpH*, *atpI* |
| | RubisCo large subunit | *rbcL* |
| Other genes | Maturase K | *matK*,*cemA* |
| | C-type cytochrome synthesis gene | *ccsA* |
| | Protease | *clpP* |
| | Proteins of unknown function | *ycf1²*, *ycf2²*, *ycf68 ²* |
| | pseudogene | *[f]ycf68²* (in *Z. odorata* ) |

**Notes.**
[2]Two gene copies in IRs.
[a]shows only one copy in *Z. rehmannii*.
[b]shows only one copy in *Z. aethiopica*.
[c]shows only one copy in *Z. odorata*.
[d]shows gene not exist in *Z. aethiopica*.
[e]shows gene not exist in *Z. odorata*.
[f]shows pseudogenes.

The level of sequence divergence in the aligned cp genome sequences from our four study species was explored using nucleotide variability ($\pi$), calculated using DnaSP version 5.1. The nucleotide variability ($\pi$) of these sequences was 0.0487, suggesting that the divergence between the cp genomes of these closely related species was relatively large. A total of 12,958 SNPs (including indels) were found. We infer from these results that the *Zantedeschia* cp genome could be suitable for species-level phylogenetic analyses.

## IR contraction and expansion in the *Zantedeschia* cp genome

A detailed comparison of the IRs of *Zantedeschia* cp genome was conducted and is presented in Fig. 3. The cp genome sequences of 11 other species of Araceae downloaded

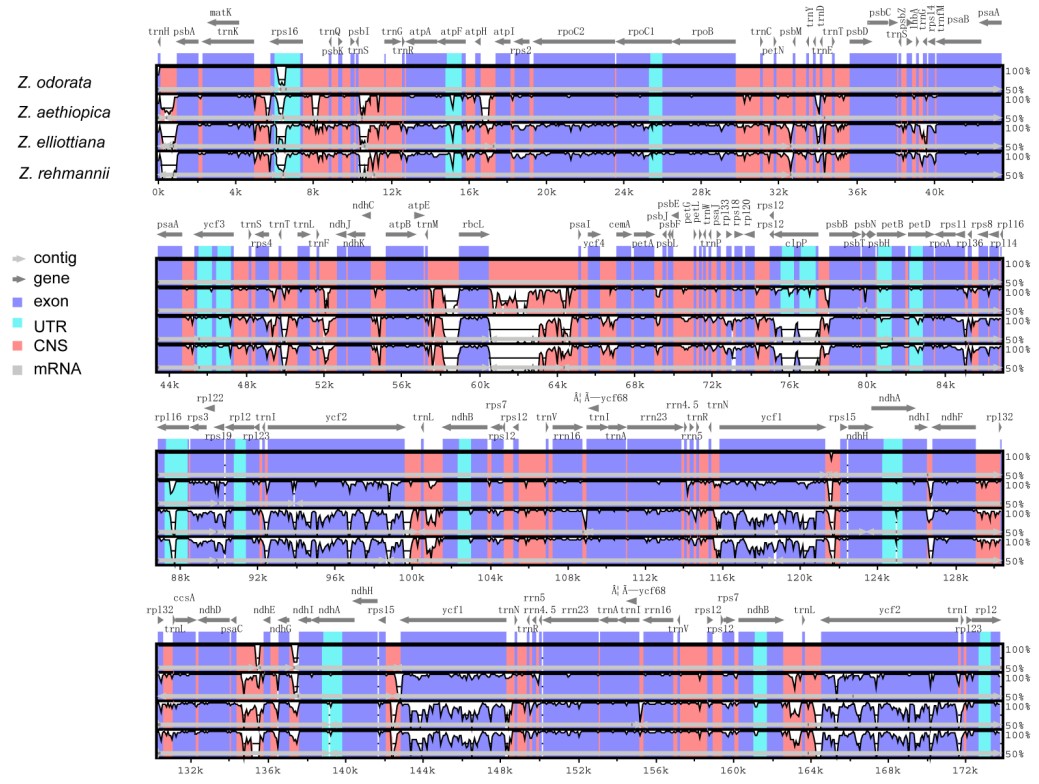

**Figure 2  Comparison of four cp genomes using the mVISTA alignment program.** The *x*-axis represents the coordinates in the cp genome. The *y*-axis indicated the average percent identity of sequence similarity in the aligned regions, ranging between 50% and 100%, Purple bars represent exons, blue bars represent untranslated regions (UTRs), pink bars represent noncoding sequences (CNS), gray bars represent mRNA, and white bars represent differences of genomics.

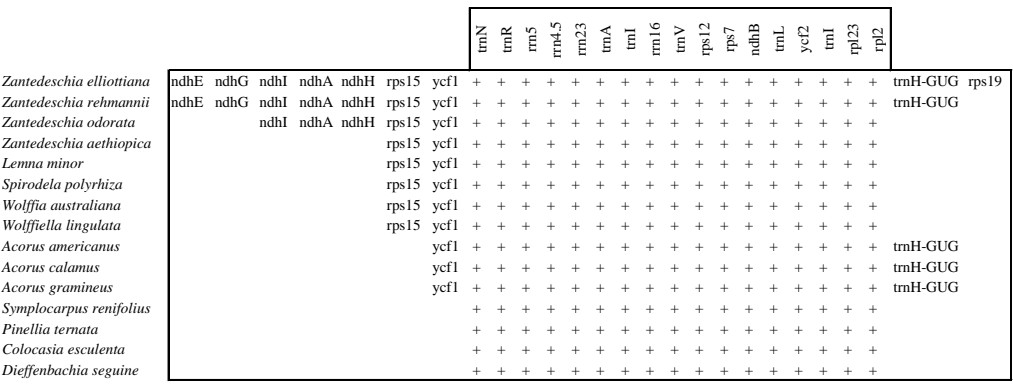

**Figure 3  Genes of IR region in Araceae (genes which are only partially duplicated in the IR are not shown).**

from NCBI were included in our analysis in order to investigate changes in the IR sequence in *Zantedeschia*.

The IR regions of *Z. aethiopica, Z. odorata, Z. elliottiana* and *Z. rehmannii* had lengths of 32,331 bp, 36,549 bp, 39,445 bp, and 38,354 bp, respectively. The even the shortest IR region of the four study species, that of *Z. aethiopica,* was longer than any of those from other species in the Araceae included in our study: *Colocasia esculenta* (25,273 bp), *Pinellia ternata* (25,625 bp), and *Dieffenbachia seguine* (25,235 bp) (*Tian et al., 2018*).This expansion of the IR in the *Z. aethiopica* cp genome is because in this species, the rps15 gene has shifted from the SSC region to IRb at the SSC/IRb border, as well as to IRa at the SSC/Ira border. Other unusual, large expansions at the borders of IR regions have also been observed in our other three study species of *Zantedeschia*. In the two species *Z. elliottiana* and *Z. rehmannii,* the SSC/IRb border of occurs beside the *ndhE* gene, meaning that six genes (*ndhE, ndhG, ndhI, ndhA, ndhH, rps15*) have shifted from the SSC region to the IR region in these species. In the case of *Z. odorata*, the SSC/IRb border occurs beside the *ndhI* gene, and four genes (*ndhI, ndhA, ndhH,* and *rps15*) have therefore shifted from the SSC to the IR region. In all cases these shifts have resulted in a large expansion of the IR region.

## Phylogenetic analysis

The cp genome sequences of 12 species (11 from the Araceae and the outgroup, *Zea mays*) were downloaded from NCBI, and the sequences were aligned together with the four *Zantedeschia* cp sequences from this study using Geneious 8.1 (*Kearse et al., 2012*). This alignment of the concatenated nucleotide sequences of a total of 16 cp genome sequences (an ingroup of 15 species from Araceae and the ougroup, *Zea mays*) was subjected to phylogenetic analyses. The phylogeny was reconstructed using maximum likelihood (ML), and the resulting phylogenetic tree was found to be in agreement with the traditional genus-level morphological taxonomy of the Araceae (Fig. 4). Furthermore, the topology is consistent with the classical taxonomy of *Zantedeschia* at the genus level (*Singh, Wyk & Baijnath, 1996*). Our four study species from the genus *Zantedeschia* formed a monophyletic clade with 100% bootstrap support. The morphologically defined sections *Zantedeschia* and *Aestivae* are also supported, with the two species from section *Zantedeschia* (*Z. odorata* and *Z. aethiopica*) sharing a more recent ancestor, and this clade forming a sister to the two species (*Z. elliottiana*, *Z. rehmannii*) from section *Aestivae*. This is the first time the cp genomes of these four *Zantedeschia* species have been sequenced, and the sequences have enriched the phylogenetic research in Araceae and we believe that they will provide a useful resource for the further study of the genetic diversity in this family.

## Simple Sequence Repeats (SSRs)

The SSR survey of the four *Zantedeschia* species in this study identified 73, 107, 110, and 120 potential SSRs motifs in the cp genome sequences of *Z. odorata* (175,906 bp), *Z. elliottiana* (175,067 bp); *Z. aethiopica* (169,065 bp), and *Z. rehmannii* (173,783 bp), respectively. The observed frequency of SSRs motifs was therefore approximately one SSR motif per 1,400-2,500 bp of cp genome (Table S2). The majority of identified SSRs were mononucleotide repeats (*Z. elliottiana*: 52; *Z. rehmannii*: 55; *Z. aethiopica*: 61; *Z. odorata*:

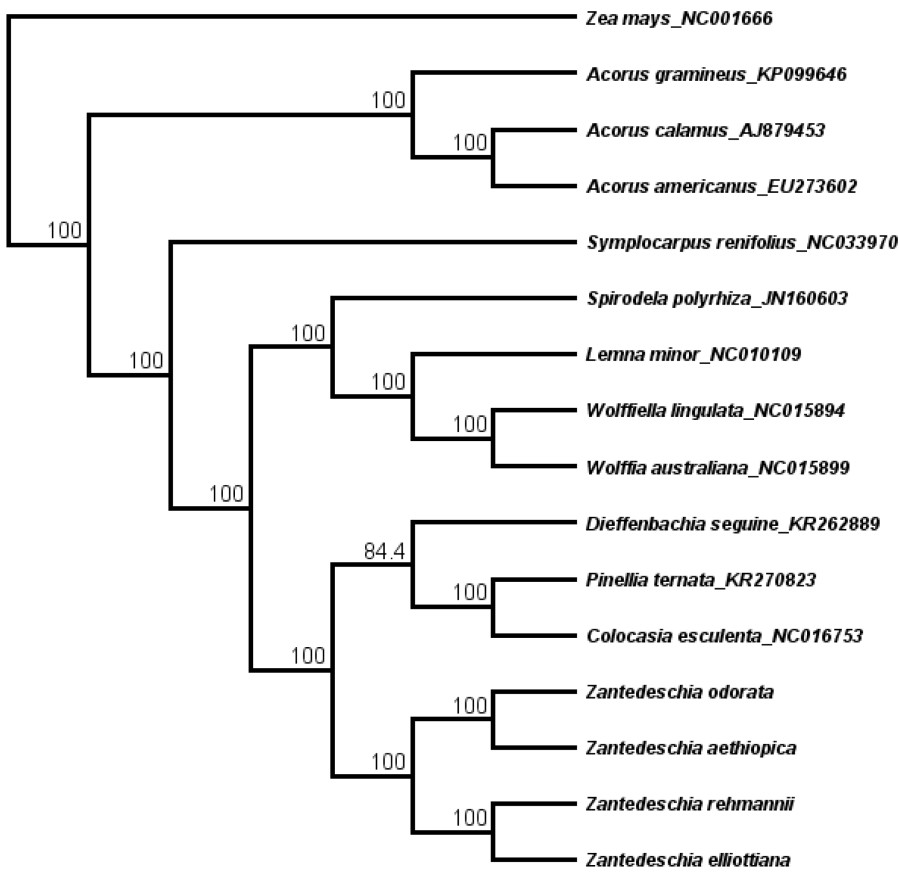

**Figure 4** **The maximum likelihood (ML) phylogenetic tree based on 14 complete chloroplast genome sequence.** Numbers at the right of nodes are bootstrap support values.

54), followed by dinucleotide repeats (*Z. elliottiana*: 32; *Z. rehmannii*: 31; *Z. aethiopica*: 20; *Z. odorata*: 14). Most SSR repeats were AT-rich, and only 38 SSR repeats in *Zantedeschia* contained cytosine. These results are consistent with previous findings that chloroplast SSRs usually consist of short polyA/T repeats (*Nguyen, Kim & Kim, 2015*). Most SSRs motifs were located in non-coding regions, in particular in the LSC region (70.0%), or in the IR regions (22.2%). Very few SSRs were located in the SSC region, and the ratio was less than 0.08%. A similar result has been observed in other studies, suggesting that the cp genome LSC region always contains high ratio of SSR motifs (*Chi et al., 2018*; *Jian et al., 2018*). Four tri-SSRs motifs (Table 3) were used to investigate genetic diversity in *Zantedeschia*. We sampled a total of 134 cultivars. All four cp SSR loci studied were polymorphic in the genus *Zantedeschia*. The number of alleles (*Na*) of the genus was 3.000, the number of effective alleles (*Ne*) was 2.371, the Shannon's information index (*I*) was 0.934 and the polymorphism information content (PIC) was 0.388 (Table 4). These results suggest that chloroplast SSR markers could be useful tools to study genetic diversity in *Zantedeschia*, and furthermore that this could be an effective method to select germplasm for the improvement of ornamental cultivars in *Zantedeschia*.

**Table 3** **Characteristics of the four SSR motifs for *Z. odorata*.** Forward and reverse primer sequences, Annealing temperature (Tm), repeat motifs.

| Primer | repeat | Start(bp) | End(bp) | Forword Primer (5′–3′) | Tm(°C) | Reverse Primer (5′–3′) | Tm(°C) |
|---|---|---|---|---|---|---|---|
| 1 | (A)10 | 4957 | 4966 | CATAGCCGCACTTAAAAGCC | 59.875 | TGGGATCGTGCAATCAATTT | 61.239 |
| 2 | (A)10 | 12561 | 12570 | CCATAAAGGAGCCGAATGAA | 60.031 | AGACAATGGACGCTGCTTTT | 59.882 |
| 3 | (A)10 | 40167 | 40176 | ATCCCCTTCTCCATCGAAAT | 59.728 | AGCAAGATTGGTTGGATTGG | 59.933 |
| 4 | (T)10 | 76955 | 76964 | GGGCAAATTATGTCAGTGCC | 60.339 | AGGCTATCTCAAACTGCCGA | 59.978 |

**Table 4** **Genetic diversity parameters estimated on 134 *Zantedeschia* accessions.**

| Parameter | Section Zantedeschia | Section Aestivae | total |
|---|---|---|---|
| *Na* | 3.000 | 14.000 | 14.000 |
| *Na* Freq. ≥ 5% | 3.000 | 6.500 | 7.000 |
| No. Private Alleles | 0.000 | 11.000 | 14.000 |
| *Ne* | 2.295 | 6.084 | 6.307 |
| *I* | 0.844 | 2.116 | 2.130 |

**Notes.**
Na, No. of Alleles; Na (Freq ≥ 5%), No. of Different Alleles with a Frequency ≥ 5%; Ne, No. of Effective Alleles; I, Shannon's Information Index.

## DISCUSSION

### IR contraction and expansion in the cp genome of the genus Zantedeschia

With the exception of certain plants in the Fabaceae, and all conifers, the cp genomes of most plants display large inverted repeats (*Aii et al., 1997*). It has been suggested that these IRs have important roles in conserving essential genes and stabilizing the structure of chloroplast DNA (*Palmer & Thompson, 1982*).Most plant species have an IR of about 25 kbp in size (*Aii et al., 1997*), and while the sequences of IRs are generally conserved, contraction and expansion events at the borders of these regions are common. During land plant evolution, there have been multiple instances of IR expansion or contraction that have involved the shifting of complete genes from the SSC regions into the IR or *vice versa*, resulting in the IR in land plants varying in size from 10 to 76 kbp. These events change the boundaries of the IR regions with the LSC or SSC regions, explaining the variation in size of the cp genome (*Raubeson et al., 2007*; *Xia, Wang & Smith, 2009*; *Yao et al., 2015*). The terminal IR gene, which is adjacent to the SSC region, is highly conserved across most land plants, and in most species,including those in the genera *Rosa*, *Lancea* and *Paeonia* (*Meng et al., 2018*),*trnN-GUU* is the last full-length IR gene at the IR/SSC boundary. This is strong evidence that this was an ancestral IR/SSC endpoint that has been retained in most lineages (*Zhu et al., 2016*).

When we compared the cp genome IR boundary in various genera in the Araceae (*Lemna*, *Symplocarpus*, *Wolffia*, *Acorus*, *Symplocarpus*, *Pinellia*, *Colocasia* and *Dieffenbachia*), we found in this family, the IR generally terminates at the *trnN-GUU* gene at the IR/SSC boundary (Fig. 3) like most land plants. Howerver, we did find several minor IR extensions into the SSC in the Araceae genera *Acorus*, *Wolffiella*, *Spirodela*, *Lemna* as well as in

*Z. aethiopica.* Minor IR extensions into the LSC region have occasionally occurred, as in *Acorus americanus.* Surprisingly, large expansions have occurred in three species of *Zantedeschia,* and in particular in *Z. elliottiana,* in which six genes of the SSC region and two genes of the LSC region have shifted into the IR region. The variation in the size of the IR region may not only explain the differences in length between different *Zantedeschia* cp genomes, but may also affect the rates of substitution and of plastome sequence evolution. Indeed, there is evidence that the IR has significant influence on the rates of evolution of plastid genomes, and the IR has been demonstrated to have lower rates of substitution (non-coding as well as synonymous and nonsynonymous) than do single-copy genes in several groups of angiosperms including carnivorous plants (*Kim, Park & Kim, 2009*; *Susann et al., 2013*; *Wolfe, Li & Sharp, 1987*; *Yi et al., 2012*) as well as in some gymnosperms, such as *Cycas* (*Wu & Chaw, 2015*). However, in plants totally lacking the IR, such as the legume clade, those genes which in other groups are IR genes have a synonymous substitution rate similar to that of single-copy genes (*Perry & Wolfe, 2002*). *Wolfe, Li & Sharp (1987)* and *Perry & Wolfe (2002)* suggest that a copy-dependent repair mechanism, such as gene conversion, would explain the lower rate of substitutions seen in the IR. Gene conversion has been demonstrated in plastids (*Khakhlova & Bock, 2006*), and is suggested to have been responsible for small increases and decreases in the size of the IR region (*Goulding et al., 1996*).

## SSR genetic diversity in Zantedeschia

Genetic diversity assessment is used to characterize germplasm and also has a role in conservation, allowing the identification of potential parents for breeding programs (*Friedt et al., 2007*). Genetic diversity in germplasm collections is commonly assessed through the use of molecular markers. Inter-sample sequence repeats (ISSRs), amplified fragment length polymorphisms (AFLPs), and random amplified polymorphic DNA (RAPD) markers have allowed the development of DNA fingerprinting for the identification of cultivars of *Zantedeschia* (*Bo et al., 2012*; *Hamada & Hagimori, 1996*), revealing levels of genetic variation (*Zhang, 2009*; *Zhen & XU, 2013*). However, as well as being labor-intensive and having only unstable reproducibility, a major weakness of these molecular markers is that they are dominant markers, and cannot therefore distinguish heterozygous and homozygous genotypes (*Tan et al., 2012*). Simple sequence repeat (SSR) markers possess several advantages over the other molecular markers, including co-dominance, high polymorphism, and good reproducibility (*Morgante, Hanafey & Powell, 2002*). Furthermore, SSRs from chloroplast DNA are powerful tools in evolutionary and population genetics (*Dong et al., 2013*; *Dong et al., 2016*; *Flannery et al., 2006*; *Suo et al., 2016*) for the construction of linkage maps and to inform the breeding of crop plants (*Powell et al., 1995*; *Xue, Wang & Zhou, 2012*), because they are uniparentally inherited and can be highly variable even intraspecifically.

*Wei et al. (2012)* developed 43 polymorphic SSRs loci from expressed sequence tags (ESTs) from white calla lily (section *Zantedeschia*). Moderately high levels of genetic diversity were reported from analyses of 24 wild or cultivated accessions of white calla lily. The observed/expected heterozygosity ($H_O$/$H_E$) was 0.501/0.662, respectively, and the

mean number of alleles per locus (Na) was 5.23. The PIC was found to be 0.446 (*Wei et al., 2012*). In a subsequent study into the genetic diversity of the colored calla lily(section *Aestivae*)using 31 EST-SSRs, *Wei et al. (2017)* showed that Na = 3.58; $H_O$ = 0.453; $H_E$ = 0.478, PIC = 0.26 and Ne = 2.18. Although the two studies both used EST-SSRs, evaluation of the genetic diversity revealed slight differences.

Our present study is the first to develop and employ SSR markers from the cp genome of genus *Zantedeschia*. In order to utilize these markers for the identification of cultivars, we choose four representative polymorphic tri-SSR markers and used these to assess genetic diversity in 134 cultivars of *Zantedeschia*. Compared with EST-SSRs diversity analyses from previous studies, our results show a low level of genetic diversity in *Zantedeschia*, with *Na* = 3.00, *Ne* = 2.371, and PIC = 0.388. Furthermore, cpSSRs showed lower diversity than the nSSRs. Similar results have been reported from other species using both types of SSR markers (*Pakkad, Ueno & Yoshimaru, 2008*; *Robledo-Arnuncio & Gil, 2005*; *Setsuko et al., 2007*), and reflects the low substitution rate in plant cpDNA sequences compared with that in nDNA (*Wolfe, Li & Sharp, 1987*). SSRs from mitochondrial or cp genomes have been developed in many species and have been used for the analysis of genetic diversity(*Song et al., 2014*; *Wheeler et al., 2014*), however this study represents the first time to develop the chloroplast SSR markers in Araceae. SSRs developed from the *Zantedeschia* uniparentally inherited and non-recombinant cp genome also have the advantages of nuclear SSRs, and we believe that they will be useful for genetic analysis in this horticulturally important genus.

## CONCLUSIONS

This study presents the sequenced cp genome sequences from four horticulturally important species of *Zantedeschia* (Araceae), a genus native to northeastern and southern Africa and now globally naturalized. The sequencing, assembly, annotation and comparative analyses revealed that the cp genome of *Zantedeschia* has a quadruple structure, with a gene order and GC content similar to those of typical angiosperm cp genomes. However, unusual IR expansion was found in this genus. SSR genetic diversity assessment showed that *Zantedeschia* has moderately high-level diversity. Phylogenetic analysis showed that the sampled species of the genus *Zantedeschia* formed a monophyletic clade. These sequences will enable us to assess genome-wide mutational dynamics within the family Araceae, and moreover, will facilitate investigations into gene expression and genetic variation within these ornamental species.

## ACKNOWLEDGEMENTS

We thank Dr. Andan Zhu and Dr. Shudong Zhang from the Kunming Institute of Botany, Chinese Academy of Sciences for their help in the revision of the manuscript.

### Funding

This study was supported by grants from the National Natural Science Foundation of China (grant nos. 31960610; 31660581 and 31500459). The funders had no role in study design, data collection and analysis, decision to publish, or preparation of the manuscript.

### Grant Disclosures

The following grant information was disclosed by the authors:
National Natural Science Foundation of China: 31960610, 31660581, 31500459.

### Competing Interests

The authors declare there are no competing interests.

### Author Contributions

- Shuilian He conceived and designed the experiments, authored or reviewed drafts of the paper, and approved the final draft.
- Yang Yang conceived and designed the experiments, performed the experiments, analyzed the data, prepared figures and/or tables, and approved the final draft.
- Ziwei Li, Xuejiao Wang and Yanbing Guo performed the experiments, analyzed the data, prepared figures and/or tables, and approved the final draft.
- Hongzhi Wu conceived and designed the experiments, authored or reviewed drafts of the paper, and approved the final draft.

### Data Availability

Data is available at NCBI: MH743153–MH743155, MG432242.

### Supplemental Information

Supplemental information for this article can be found online at http://dx.doi.org/10.7717/peerj.9132#supplemental-information.

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
