# Peer review of "Comparative analysis of four Zantedeschia chloroplast genomes: expansion and contraction of the IR region, phylogenetic analyses and SSR genetic diversity assessment"

_PeerJ, doi:10.7717/peerj.9132_

## Round 0.1 · original submission · Major Revisions

Your manuscript has been evaluated by three reviewers (see below). Reviewers and myself agree that this study is interesting as it will supply more genomic information for many different researchers. However, I do not think that the manuscript is quite ready for publication in its present form and therefore suggest reconsideration of the manuscript after revision.

The methodology is obscure at some points and it will need some clarification. The source of sequenced plants (also for SSR genotyping), number of accessions, etc. The number of plants of each section in the SSR analysis seems very unbalanced (7 vs. 126). Author should try a better sampling. In addition, I agree reviewer #2 that the use of SSRs is timely but mononucleotide repeats should be avoided. They are more prone to errors in PCR amplification. Authors should use di, tri or tetra SSRs.

Regarding cp genome assembly, authors should provide the gene list with appropriate annotations (e.g. GO:). In addition, as outgroup the authors chose not the available (core) Monocot (e.g. Poaceae) but the highly evolved, Arabidopsis thaliana, a distant (evolutionary speaking) relative, inviting ingroup-outgroup branching artefacts. The accession numbers of Araceae genera should be provided. The number of cp genomes (14) doesn’t fit the number of accessions in Figure 7 (13).
Discussion should be improved and largely re-written. IR dynamic variation should be effectively discussed. The SSR section should be improved following the new SSR data.

Lastly, the English in the present manuscript is not of publication quality and require major improvement. Please carefully proof-read and spell check to eliminate grammatical errors and to correct many typos (e.g. L262,268). When submitting your revised paper, please provide a cover letter to explain, point by point, the details of the revisions in the manuscript and your responses to the referees' comments.

·

Basic reporting

yes

Experimental design

yes

Validity of the findings

yes

Additional comments

These cp genomes were well assembled (de novo) and the variations were also fully examined, so I support publication of this study.

Reviewer 2 ·

Basic reporting

The whole English expression is fair, some places need to improve. Please do more polishings work on this.

the background about the mechanism and meaning of SSRs is not enough in the introduction parts.

The figures of four cp genome are high repeated. And they will not supply more useful information for the readers.

Experimental design

The whole design is good, And the extra SSR markers are used to re-test the meaning of them are nice for readers.

Validity of the findings

Sequencing the complete cp genomes have become more and more easy and popular in land plants. However, the whole analysis is still focusing on data reporting. Except the SSRs part, all other findings are repeated from previous reports or the same as them. There are no exciting findings from the cp genome work.

Additional comments

Dear authors,

The manuscript from He et al. as "Comparative analysis of four Zantedeschia chloroplast genomes: expansion and contraction of the IR region, phylogenetic analyses and SSR genetic diversity assessment". In this paper, the authors finished four full cp genome in Zantedeschia genus by using the NGS method and did comparisons analysis across many different species. It is always good to get the genome data from different plant groups. That will supply more genomic information for many different researchers. But the whole analysis from this ms is not exciting because all parts are following the same styles from many other plant cp genome paper as data reporting. The whole MS is data descriptions and comparisons. And the only good point is deep into the SSR analysis with more populations data. In all, it is will be nice as the data reporting paper, not as the research paper.

Major parts:
Introduction:
As many papers reported, the cp genomes have been published from thousands of plant. The whole research style is almost the same as each other. From the introduction, methods, results and discussion, all parts are following the same pattern. We can not get any valuable scientific view after the reading. If the whole design is as data reporting, it should be fine.
The authors used the SSR in their title, then the SSR should be the valuable point to deep into. The introduction part should extend this part to interest the readers.
It is also nice to extend the point about using the cp genome as a marker to explore the phylogeny and barcoding of this confused groups for hybridization and all kinds of breeding.
Discussion:
IR dynamic variation is really nice to discuss, but in this ms, the authors just put the number here. we did not see any mechanism about it.
The mechanism of SSR and utility of them should be discussed.

Minor point
1. please make it uniform for using the "cp" and "chloroplast"

2. please add the source of those plants. where and how you get them? how to plant them in the lab?

3. the CTAB method is used to extract the total DNA. it is confused to say "Total genomic DNA 49 was isolated from fresh leaves of these seedlings, using a modified cetyltrimethyl ammonium 50 bromide (CTAB) method (Doyle 1987) to construct chloroplast DNA libraries." It seems as cp DNA libraries.

4 "Insertion/deletion polymorphisms (indels) were identified using DnaSP", how did you do the alignment at here?

5. the sampling analysis of SSR should be in the method part, you put them as in the results part.

6. line 147 to 152: the divergent was relatively large? it should be compared with some markers. what kinds of data do you use to cmpare?

7. the SSR from mononucleotide repeats is also more like as the error from DNA replication. this may be not good to use as markers.

8. English writing in someplace should be improved.

·

Basic reporting

There is no problem with the structure, figures, tables and reference citations of this article. I could only suggest that the authors may probably like to improve some of their English statements of the 'Introduction' part.

Experimental design

The experiments and analysis are very standard.

Validity of the findings

All findings and conclusions are precise and well supported by the corresponding data in the paper.

Additional comments

Interesting little article, congratulations!

---

## Round 0.2 · Minor Revisions

The reviewers have recommended publication. There are a few lingering problems with use of language, which I highlighted in the attached manuscript with my own edits and comments. After these recommended, minor revisions, the manuscript will be acceptable for publication. Please refer to the attached file with my edits.

Reviewer 2 ·

Basic reporting

good

Experimental design

finished

Validity of the findings

good

Additional comments

no

·

Basic reporting

no comment

Experimental design

no comment

Validity of the findings

no comment

Additional comments

no comment

---

## Round 0.3 · accepted · Accept

The authors have addressed all remaining concerns regarding use of language, thus the manuscript is now ready for acceptance. Congratulations!